# Parenting Styles, Food Parenting Practices, Family Meals, and Weight Status of African American Families

**DOI:** 10.3390/ijerph20021382

**Published:** 2023-01-12

**Authors:** Azam Ardakani, Lillie Monroe-Lord, Dorothy Wakefield, Chimene Castor

**Affiliations:** 1Department of Nutritional Sciences, Howard University, Washington, DC 20059, USA; 2Center for Nutrition, Diet and Health, University of the District of Columbia, Washington, DC 20008, USA

**Keywords:** parenting styles, food parenting practices, family meal, obesity, African American, parent–adolescent dyads

## Abstract

Parents influence adolescents’ weight status through different strategies used in the home environment, including parenting styles (PSs), food parenting practices (FPPs), and family meal frequency. As the prevalence of obesity is higher among African American adolescents, investigation of which parental strategies serve as an adjustable factor for the prevention of obesity is critical. First, this study aims to examine the relationship between the different parenting influences and obesity statuses of both parents and 10–17-year-old adolescents among African American families. Second, it aims to examine the correlation between PSs and FPPs and frequency of family meals. A total of 211 parent–adolescent dyads completed an online survey using Qualtrics. Four PSs (i.e., authoritative, authoritarian, setting rules/expectations, and neglecting) and four FPPs (i.e., monitoring, reasoning, copying, and modeling) were identified for this study, along with family meal frequency. Body mass index (BMI) percentile and BMI were used to assess the obesity status of the adolescents and parents, respectively. No correlation was found between the adolescents’ and parents’ obesity status and the PSs and FPPs, while the adolescents’ BMI percentile was significantly correlated with parental BMI. However, a higher number of family meals decreased the likelihood of obesity among the adolescents to some extend and depended on the type of BMI used. An authoritative PS was the only style related to family meal frequency, while three FPPs, namely, monitoring, reasoning, and modeling, were related to a greater number of family meals in African American families. The findings of this study can be used in the development of parental education workshops/sessions, with consideration of the cultural differences in African American families, and can help parents to adopt the best parenting strategy to promote the healthy weight status of their adolescents.

## 1. Introduction

Obesity is a major public health issue and a growing concern universally. The obesity prevalence is around 21% among older adolescents aged 12–19 years, which is higher than that of younger children and adolescents [1]. The National Center for Health Statistics reported that the obesity prevalence is higher among African American youth in comparison to White youth [2]. Moreover, overweight and obese children and adolescents are more likely to be overweight and develop a chronic disease in adulthood [3,4].

Environmental, behavioral, and personal factors exert an influence on eating habits and behaviors as an element involved in developing or preventing obesity in children and adolescents [5]. Lifestyle choices, psychological factors, family factors, and socioeconomic factors are the most remarkable etiologies for childhood obesity [6]. Parenting (or caregiver) styles (PSs), food parenting practices (FPPs), and frequency of family meals are major factors among different environmental factors that impact children and adolescents as the first and most influential community that they join. PSs refer to the engagement and responsiveness level of parents in different situations with their child. FPPs are postulated to impact children’s eating behaviors [7,8]. FPPs were also identified as a predictor of children and adolescents’ health outcomes in adulthood [9].

A high frequency of family meals provides several benefits for families, which include improving weight status and promoting healthy eating habits [10,11]. An absence of family meals is associated with unhealthy eating patterns and poor diet quality [12]. Furthermore, there is a negative association between the frequency of family meals and obesity development [13]. However, the question of how eating meals together as a family is related to other aspects of the family environment, such as different PSs and FPPs, and whether this relationship is associated with obesity status among African American minority groups is unresolved.

African American parents were shown to use an authoritarian PS, which is characterized by high restriction and monitoring of children’s food consumption [14]. Many African American children and adolescents do not meet the recommended dietary intake of fruits, vegetables, and whole grains due to low socioeconomic status, which can lead to a higher risk of obesity [15].

Family system theory (FST) was used as the theoretical framework for this study. This theory emphasizes the importance of the family as a system to understand and explain individual behaviors in the context of family interactions [16]. FST suggests that any change in family structure or the role of family members can have an impact on the behavior of the entire family over time [17]. Previous studies revealed that a warm and supportive PS correlates with the number of desirable healthy behaviors practiced. This can impact adolescent weight status and dietary patterns [18,19,20]. In contrast, restrictive FPPs, such as pressure to eat, restrictions on youth’s access to foods, and parental concerns about adolescents’ weight status, are associated with poorer diet quality [21,22].

There are very few studies addressing the impact of the family environment on the obesity status of both parents and adolescents, especially among minorities. The importance of this work is to study the effect of three influential factors of family environment, including PSs, FPPs, and family meal frequency together on obesity status among African American families. This study becomes even more important considering that obesity is a major problem among adolescents of minority groups, especially African Americans. The goal of this study is to help elucidate which family environmental factors have a positive impact on controlling the weight status of African American families and communities, and to determine which PSs and/or FPPs may lead to a higher family meal frequency. The results indicate higher family meal frequency with positive correlation with healthier weight status among African American adolescents, and authoritative PS and monitoring, reasoning, and modeling FPPs with higher frequency of family meals.

## 2. Methods

### 2.1. Research Design, Participants, and Procedure

The protocol of the current study was approved by the institutional review board (IRB) of the University of the District of Columbia. A total of 211 African American parent–adolescent dyads participated in this cross-sectional study. The dyads were recruited by Qualtrics from November to December 2021 to complete the survey. The inclusion criteria included the following: parents or caregivers willing to participate in the study with their 10–17-year-old adolescents; access to the Internet; being comfortable reading and writing in English; being responsible for providing food for the adolescent. All participants signed a parental consent or adolescent assent form before participating as a prerequisite for the survey.

### 2.2. Parents’ Survey

The parents completed a 20–25 min online survey. The survey used questions from the 85-item Comprehensive General Parenting Questionnaire (CGPQ), which facilitate research exploring how parenting impacts a child’s weight-related behaviors and items used by a Monroe-Lord et al. (2021) study on African American families [23,24]. The demographic characteristics of both the adolescents and the parents, household food security, household acculturation, and participation in federal food assistance programs were evaluated. The parents self-reported anthropometric measurements including height and weight for themselves and their children. Body mass index (BMI) for parents and BMI percentile for adolescents were used in this study. BMI was calculated with participants’ weight divided by the square of height used for parents. As BMI increases with age during childhood and adolescence, and it is different between males and female, BMI-for-age percentile based on CDC growth charts were used for obesity status for adolescents. BMI was categorized into three groups, including normal weight if BMI was between 18.5 and 24.9, overweight if BMI was 25.0–29.9, and obesity if BMI was 30.0 and above. BMI percentile also was categorized into three groups, including normal weight if the BMI percentile was equal to or greater than the 5th percentile and less than 85th percentile, overweight if BMI percentile was at or the 85th percentile but less than the 95th percentile, and obesity if BMI percentile was at or above the 95th percentile for specific age, gender, and height [25,26]. The survey also included the following question: “During the past 7 days, how many times did all, or most, of your family in your house eat a meal together?” The answer included six options, from “Never” to “More than 7 times.” For this study, frequency of family meals was categorized into three groups after combining the answer choices, namely, two times or less, three to six times, and 7seven times or more [27].

### 2.3. Statistical Analysis

Two exploratory factor analyses were run to identify the PSs and FPPs. Once the factors were identified, average factor scores for each parent were calculated. Spearman’s rank correlation (when weight status was considered as a continuous variable) and the Wilcoxon rank sum test (when weight status was considered as a categorical variable) were used to test the relationship between BMI percentile and BMI for both the adolescents and the parents, respectively, and PSs and FPPs. The Wilcoxon rank sum test was used to examine the relationships between family meal frequency and weight status, as well as PSs and FPPs. Spearman’s correlation was used to test the relationship between adolescent BMI percentile and parental BMI. SAS 9.4 (SAS Institute, Cary, NC, USA) was used for statistical analysis in this study. The results are considered significant at *p* < 0.05.

## 3. Results

### 3.1. Demographic Analysis

The details of the sample characteristics are presented in Table 1. The adolescent sample was composed of 41% male and 59% female individuals with a mean age of 14.28 years. The mean BMI percentile was 71.35. The obesity rate of the adolescents was 19.6% (the national estimate for African American youth is 22%). Approximately 82% of the caregiver participants in this study were the parents of the adolescents (we use the term parents for them). Most of the parents were female (70%), and 57% of the parents were overweight or obese. Approximately 56% of the parents had a college education or above. Furthermore, approximately 52% of the adolescents lived in single-parent households, and more than half of the families had family meals three to six times per week.

### 3.2. Parenting Styles and Food Parenting Practices

Two exploratory factor analyses were run for sets of 35 (for PSs) and 33 (for FPPs) items. The first factor analysis for the identification of PSs produced four factors, which were named authoritative, authoritarian, setting rules/expectations, and neglecting. One item was excluded because it did not load with any of the other four factors. The items for each PS are listed in Table 2.

A second factor analysis was run for the identification of FPPs, at which point five items were excluded from the final factors. The analysis produced four factors, which were named monitoring, reasoning, copying, and role modeling. Monitoring is defined as parents keeping track of what and how much their children eat. Reasoning or teaching is defined as parents reasoning with the child about the benefits of healthy food and teaching them healthy eating habits. Copying is defined as when parents intentionally or unintentionally encourage the child to copy their eating behaviors. Role modeling is defined as parents exhibiting healthy eating behaviors to encourage similar behaviors in their children. The items for each FPP factor are listed in Table 3.

The factor loadings and the details of the factor analyses for each PS and FPP are shown in Table 4. The internal reliability of each factor was good (Cronbach’s alpha > 0.8) or acceptable (Cronbach’s alpha > 0.7) for all PSs and FPPs. The parents received a score on all eight factors.

The highest median scores were for the authoritative and setting rules PSs. Setting rules, authoritative, neglecting, and authoritarian were the PSs applied by African American parents most prevalently and, respectively, while role modeling, copying, reasoning, and monitoring were used most prevalently and, respectively, as FPPs.

### 3.3. Relationship of Different Demographic Data with Weight Status of Both Adolescents and Parents

The relationship between the weight status of both parents and adolescents and different demographic variables were examined and reported in Table 5. Based on the results, both adolescent and parent sex were meaningfully related to BMI percentile of adolescent (*p* = 0.012 and *p* = 0.0485, respectively). Fewer male adolescents (40.6%) were in the normal weight group compared with female adolescents (63.5%), and adolescents whose main caregiver was female were in the better weight status compared with those whose caregiver was male. Male parents significantly had higher BMI compared to female parents (*p* = 0.0081). In addition, there was a significant trend toward lower BMI among the younger parents (*p* = 0.0286). Interestingly, we could not find any relationship between socioeconomic factors, including parental education and household income, and obesity status of both parent and adolescents of African American.

### 3.4. Relationship of Parent Weight Status with Adolesscent Weight Status

The relationship between parent BMI and adolescent BMI percentiles was examined. The relationship between parent BMI and adolescent BMI percentiles was examined. A highly significant correlation between parent BMI and adolescent BMI percentile was found (r = 0.42, *p* < 0.0001).

### 3.5. Relationship of PSs, FPPs, and Family Meal Frequency with Adolescents’ Weight Status

BMI percentile was considered a categorical variable (normal weight, overweight, and obese) to evaluate whether the PSs and FPPs are correlated with the obesity status of the African American adolescents. No meaningful relationship was found between the categorized adolescents’ BMI percentiles and PSs and FPPs (Table 6). In addition, no correlation was found between parent’s weight status and PSs, FPPs, and family meal.

Family meal frequency was associated with the adolescents’ BMI percentile (*p =* 0.03). The median BMI percentile score was 87.06, which indicates overweight, for those adolescents with two or fewer family meals, while it was 62.45, which indicates a normal weight, for those adolescents with more than seven family meals per week. Although no significant correlation was found between parental BMI and family meal frequency (*p =* 0.33), there was a positive trend, with a decrease in parental BMI when having more family meals (Table 7).

### 3.6. Relationship of Family Meal Frequency with PSs and FPPs

Among different studied PSs, only the authoritative PS was positively related to family meal frequency (*p =* 0.0004). The authoritative score was one score higher in those families with seven or more family meals compared to those with two or fewer family meals. However, among the four different FPPs, three of them—monitoring, reasoning, and modeling—were correlated with the frequency of family meals (*p =* 0.0002, *p =* 0.0017, and *p =* 0.0008, respectively) (Table 8).

## 4. Discussion

In this study, the relationship between different parental influences (i.e., PSs, FPPs, and family meals) and African American families’ obesity status was evaluated. Notably, the existing literature examining the influence of PSs and FPPs on obesity status among both parents and adolescents in a minority population, specifically African American, is sparse. The findings of this study reveal that African American families establish set rules and expectations more than other PSs, and authoritarian was the least prevalent PS. A previous study revealed that PS characterized by rigidity, restriction, and high control, which are classed as authoritarian styles, is more prevalent among African Americans [28]. This type of PS evokes a sense of safety and nurturance among adolescents [29]. Setting a large number of rules and expectations is a form of behavioral control by parents, which can be perceived as an authoritarian style by adolescents. Thus, it can play a negative role in health behaviors among adolescents, instead of resulting in improvements in their health status. It is important to have a supportive and alternative plan for adolescents while establishing rules and expectations. Moreover, role modeling was the dominant FPP among the African American adolescents in this study, while monitoring was the least prevalent. This finding is consistent with a previous study with a small sample size that claimed a higher score for role modeling compared to other FPPs among African American families [30].

This study could not find any relationship between the BMI percentiles/BMI of the adolescents/parents and PSs and FPPs. This finding is consistent with two recent studies and one older study that reported no specific correlation between FPPs and being overweight or obesity in children and adolescents [2,30,31,32,33]. In addition, two other studies confirmed that maternal weight status is an independent factor of FPPs [21,24]. However, a previous study showed that greater parental responsiveness, which is characteristic of an authoritative PS, is significantly correlated with a lower BMI percentile [34]. These different findings regarding the correlation between authoritative PSs and obesity status could result from the different questionnaire design used in the two studies. It is important to note that although we could not find any statistically meaningful correlation, the association between authoritative PSs and the obesity status of the adolescents was negative. It is important to consider the impact of PSs or FPPs on obesity status in adulthood. Its impact can be highlighted in the future of adolescents.

Family meal frequency was associated with the weight status of the adolescents. The adolescents who had more family meals per week had a lower BMI percentile. Although no significant correlation was found between parental BMI and family meal frequency, a positive impact on the parents’ obesity status was observed, as also those adolescents with three or more family meals per week were in the normal weight status group, compared to those who had two or fewer family meals who were overweight. Previous findings also showed that family meal frequency can control the development of obesity among children and adolescents [13]. In addition, family meals during adolescence not only maintain adolescents’ normal weight status, but also help to protect them from the development of becoming overweight and obese in young adulthood [35]. This is due to learning how to choose healthier and more nutrient-dense foods during family meals, which impacts their dietary habits and aids in the prevention of the consumption of unhealthy snacks and foods.

Authoritative was the only PS positively associated with the frequency of family meals. Previously, a study with a small sample size of overweight and obese African American adolescents demonstrated that an authoritative style contributes to improving the family meal frequency [22]. Three out of four FPPs (i.e., monitoring, reasoning, and modeling) significantly impacted the frequency of family meals. Paying attention to FPPs can help to create positive changes in establishing healthier behaviors, such as family meals in comparison to PSs. FPPs may not only promote healthier diets among adolescents, but may also help to promote better psychological health for family members [11,36].

The strength of this study is the consideration of African American individuals as a minority group who are one of the most vulnerable populations to obesity. Future studies can use other methods, such as bio-impedance, waist circumference, and dual X-ray absorptiometry to examine the correlation between obesity status and parental influences. The majority of the previous studies on PSs, FPPs, family meals, and adolescents’ weight status focused on one of the parent or caregiver variables, especially maternal influences, in minority groups such as African American families. Future studies can focus on the father’s styles and practices in terms of the weight status and dietary habits of adolescents. In addition, intervention studies can also help us to understand which and how different parental influences (i.e., PSs, FPPs, and family meal frequency) can be most useful in maintaining a normal weight status while considering the cultural values of minority groups.

## 5. Conclusions

This study focused on different parental influences, including PSs, FPPs, and family meal frequency, and their relationships with the weight status of African American families. We also examined how each PS and FPP can impact the family meal frequency. The results indicate that family meal frequency plays a more important role in ensuring a healthy weight status among African American adolescents in comparison to PSs and FPPs. An authoritative PS was the only style correlated with a higher family meal frequency, while monitoring, reasoning, and modeling practices were correlated with a higher frequency of family meals.

## Figures and Tables

**Table 1 ijerph-20-01382-t001:** Demographic characteristics for the African American dyad participants.

Characteristics	*N*	%	Mean (SD)
Adolescent age (years)			14.28 (2.32)
10–13	82	38.86	
14–17	129	61.14	
Adolescent sex			71.35 (27.88)
Male	87	41.23	
Female	124	58.77	
Adolescent weight status (percentile)			
Normal ^1^	92	54.76	
Overweight ^2^	43	25.60	
Obese ^3^	33	19.64	
Parents’ age (years)			
18–25	42	19.91	
26–34	66	31.28	
35–54	98	46.45	
55–64	3	1.42	
≥65	2	0.95	
Parents’ sex			27.41 (6.74)
Male	63	29.86	
Female	148	70.14	
Parents’ weight status			
Normal ^4^	68	42.50	
Overweight ^5^	55	34.38	
Obese ^6^	37	23.13	
Parents’ education			
Below high school	80	37.91	
Diploma or GED	12	5.69	
Some college or technical school	56	26.54	
≥4 Years of college	63	29.86	
Household income (USD)			
<25,000	34	16.11	
25,000–44,999	51	24.17	
45,000–64,999	37	17.54	
65,000–84,999	33	15.64	
≥85,000	49	23.22	
Prefer not to answer	7	3.32	
Marital status			
Single ^7^	109	51.67	
Married	102	48.34	
Relationship with adolescent			
Parent (includes step- or foster parent)	173	81.99	
Aunt/uncle	2	0.95	
Grandparent	12	5.69	
Sibling	23	10.90	
Other	1	0.47	
Meals together			
2 or fewer times	55	26.07	
3–6 times	117	55.45	
7 or more times	39	18.48	

^1^ 5 < BMI percentile < 85; ^2^ 85 ≤ BMI percentile < 95; ^3^ BMI percentile ≥ 95; ^4^ 18.5 ≤ BMI < 25; ^5^ 25 ≤ BMI < 30; ^6^ BMI ≥ 30; ^7^ single, divorced, never married, and widowed.

**Table 2 ijerph-20-01382-t002:** Parenting styles survey items.

Parenting Styles
Authoritative
I know exactly when things are not going well for my child.
When my child is sad, I know what is going on with him or her.
I feel good about the relationship I have with my child.
My child and I have warm affectionate moments together.
I know exactly when my child has difficulty with something.
I find time to talk with my child.
I spend a lot of time with my child.
I easily find a way to make time for my child.
I attend as many of my child’s events and activities as possible.
I find it interesting and educational to be with my child for long periods of time.
Every free minute I have I spend with my child.
I always help my child with everything he/she does.
Authoritarian
I have a hard time consistently enforcing rules with my child.
There are times I just do not have the energy to make my child behave as he or she should.
When my child does something that is not allowed, I do not talk to him or her until he or she says he or she is sorry.
I am less friendly with my child if he or she does not see things my way.
I make sure my child is aware of how much I sacrifice for him or her.
I make my child feel guilty when he or she does not meet my expectation.
When my child hurts my feelings, I stop talking to him/her until he or she pleases me again.
I do not allow my child to question my decisions.
I do not allow my child to get angry with me.
When my child has lost something, I stop what I am doing to find it before he/she gets too upset.
I do not let my child get involved in activities or tasks where he/she may potentially fail.
I carefully plan my child’s day so that he/she has enough activities to keep him/her busy.
Setting Rules/Expectations
I expect my child to follow our family rules.
I have clear expectations for how my child should behave.
I require my child to behave in certain ways.
I make sure that my child understands what I expect of him or her.
I teach my child to follow rules.
When I ask my child to do something, I expect him/her to do without any questions.
I let my child know that I am the boss in our house.
Neglecting
I do not always follow through when I threaten to discipline my child.
I threaten discipline more often that I actually give it.
When I discipline my child, I sometimes end the punishment early.

**Table 3 ijerph-20-01382-t003:** Food parenting practices survey items.

Food Parenting Practices
Monitoring
How much do you keep track of the sweets (candy, ice cream, cake pastries) that your child eats?
How much do you keep track of the sugary drinks (soda/pop, Kool Aid) that your child drinks?
How much do you keep track of the snack foods (potato chips, Doritos, cheese puffs) that your child eats?
How much do you keep track of the high-fat foods (fried foods, French fries) that your child eats?
How much do you keep track of the fruits and vegetables that your child eats?
How much do you keep track of the milk or foods with calcium, like cheese and yogurt, that your child consumes?
How much do you keep track of foods labeled as whole grain that your child eats?
I like to be sure that my child does not eat too many sweets (candy, ice cream, cake, pastries).
I like to be sure that my child does not eat too many high-fat foods.
I like to be sure that my child does not eat too much of his or her favorite food.
Reasoning/Teaching
How often do you say something positive about the food that your child is eating?
How often do you tell your child how tasty a new food is?
How often do you reason with your child to get him/her to eat (for example, milk is good for your health because it will make you strong)?
How often do you tell your child that healthy food tastes good?
How often do you compliment your child for eating food (for example, “what a good boy/girl! You’re eating your vegetables”)?
How often do you encourage your child to try to eat healthy foods such as vegetables?
I explain my food choices verbally to my child (e.g., “I think I’m going to have some fruit, as I like it and it’s good for me”).
Copying
I verbally encourage my child to copy my eating behaviors.
I try to talk more often about foods I would like my child to eat.
My child picked up eating behaviors from me that I tried to hide from him or her (e.g., avoiding certain foods).
My child copied eating habits from me that I did not realize I had (e.g., salting my food before I taste it).
If I point out certain eating behaviors or foods I do or do not like, my child is more likely to copy them.
The eating behaviors of other family members influence what my child eats.
Role Modeling
My child picked up eating behaviors from me that I did not intentionally encourage him or her to copy (e.g., putting ketchup on most foods or eating vegetables first).
When I show my child I enjoy fruits and vegetables, he or she tries them.
My child is more likely to try or eat new foods if I eat the new foods with him or her.
My child is more likely to try new foods he or she saw me eating.
My child asks to try foods from my plate that he or she sees me eating.

**Table 4 ijerph-20-01382-t004:** Factor analysis to derive parenting practices.

	*N*	Factor Loadings (Min–Max)	Cronbach’s Alpha	Mean (SD)	Median (IQR)
**Parenting styles**					
Authoritative	12	0.55–0.70	0.89	4.09 (0.79)	4.25 (3.5–4.83)
Authoritarian	12	0.41–0.76	0.87	3.44 (0.94)	3.41 (2.91–4.25)
Setting rules	7	0.41–0.59	0.82	4.09 (0.82)	4.28 (3.57–4.85)
Neglecting	3	0.49–0.71	0.77	3.59 (1.15)	3.66 (3.00–4.66)
**Parenting practices**					
Monitoring	10	0.47–0.70	0.89	3.43 (0.86)	3.40 (2.93–4)
Reasoning	7	0.34–0.72	0.84	3.45 (0.87)	3.42 (2.85–4.1)
Copying	6	0.40–0.67	0.76	3.41 (0.82)	3.50 (3–4)
Modeling	5	0.38–0.70	0.78	3.62 (0.82)	3.60 (3–4.2)

**Table 5 ijerph-20-01382-t005:** Correlation between different demographic data and weigh status of both adolescents and parents.

	Adolescent Weight Status *n* (%)				Parent Weight Status			
	Normal	Overweight	Obese	*p* Value	Normal	Overweight	Obese	*p* Value
Adolescent age				0.8570				
10–13	34 (54.8)	17 (27.4)	11 (17.5)					
14–17	58 (54.7)	26 (24.5)	22 (20.8)					
Adolescent sex				0.0120				
Female	66 (63.5)	23 (22.1)	15 (14.4)					
Male	26 (40.6)	20 (31.3)	18 (28.1)					
Parent age				0.5316				0.0286
<35	49 (57.0)	23 (26.7)	14 (16.3)		42 (50.6)	26 (31.3)	15 (18.1)	
≥35	43 (52.4)	20 (24.4)	19 (23.2)		26 (33.8)	29 (37.7)	22 (28.5)	
Parent sex				0.0485				0.0081
Female	74 (58.7)	31 (24.6)	21 (16.7)		57 (46.7)	34 (27.9)	31 (25.4)	
Male	18 (42.8)	12 (28.6)	12 (28.6)		11 (28.9)	21 (55.3)	6 (15.8)	
Parent education				0.1065				0.2328
HS ^1^ or less	23 (46.0)	14 (28.0)	13 (26.0)		16 (32.6)	19 (38.8)	14 (28.6)	
More than HS	69 (58.5)	29 (24.6)	20 (16.9)		52 (46.9)	36 (32.4)	23 (20.7)	
Household income				0.9188				0.1207
Less USD 45,000	36 (52.9)	19 (27.9)	13 (19.2)		25 (36.8)	22 (32.3)	21 (30.9)	
USD 45,000 or more	52 (55.3)	23 (24.5)	19 (20.2)		42 (17.5)	29 (33.7)	15 (48.8)	
Marital status				0.0754				0.5011
Married	45 (56.9)	24 (30.4)	10 (12.7)		35 (47.3)	24 (32.4)	15 (20.3)	
Not married	47 (52.8)	19 (21.4)	23 (25.8)		33 (38.4)	31 (36.0)	22 (25.6)	
Meal together				0.0284				0.3260

^1^ High school.

**Table 6 ijerph-20-01382-t006:** Correlation between the categorized BMI percentiles of the adolescents and parenting styles and food parenting practices.

	Adolescents’ Weight Category			*p*-Value
	Normal ^1^	Overweight ^2^	Obese ^3^	
	Median (IQR) ^4^ Mean (SD)	Median (IQR) Mean (SD)	Median (IQR) Mean (SD)	
**Parenting Styles**				
Authoritative	4.33 (3.67–4.83)	4.33 (3.58–4.75)	4.25 (3.42–4.83)	0.70
	4.17 (0.72)	4.12 (0.69)	4.04 (0.81)	
Authoritarian	3.33 (2.58–4.17)	3.25 (3.0–4.08)	3.41 (3.17–4.17)	0.53
	3.33 (0.95)	3.45 (0.89)	3.52 (0.84)	
Setting rules	4.28 (3.71–4.71)	4.42 (3.71–5.0)	4.28 (3.28–4.57)	0.48
	4.14 (0.74)	4.22 (0.77)	4.03 (0.78)	
Neglecting	3.67 (3.0–4.67)	3.33 (2.67–4.33)	4.0 (3.0–4.33)	0.85
	3.59 (1.17)	3.49 (1.10)	3.58 (1.06)	
**Parenting Practices**				
Monitoring	3.40 (2.85–3.95)	3.40 (2.85–3.95)	3.30 (3.0–3.90)	0.79
	3.40 (0.91)	3.40 (0.91)	3.41 (0.76)	
Reasoning	3.28 (2.85–4.14)	3.42 (2.57–4.12)	3.42 (2.85–4.14)	0.84
	3.42 (0.84)	3.39 (0.98)	3.53 (0.80)	
Copying	3.41 (2.92–40)	3.50 (2.83–4.0)	3.33 (3.0–4.0)	0.94
	3.39 (0.82)	3.44 (0.80)	3.38 (0.75)	
Modeling	3.60 (3.0–4.20)	3.60 (3.2–4.40)	3.60 (3.20–4.20)	0.60
	3.59 (0.85)	3.79 (0.71)	3.62 (0.68)	

^1^ 5 < BMI percentile < 85; ^2^ 85 ≤ BMI percentile < 95; ^3^ BMI percentile ≥ 95; and ^4^ median (IQR) of the parenting styles and parenting practices.

**Table 7 ijerph-20-01382-t007:** Relationship between weight status of adolescents and family meal frequency.

	Number of Times per Week Families Ate Together			*p*-Value
	≤2 Times	3–6 Times	≥7 Times	
	*N* (%)	*N* (%)	*N* (%)	
**Adolescent’s Weight status**				
Normal weight	19 (46.3)	51 (53.1)	22 (71.0)	0.0284
Overweight	8 (19.5)	30 (31.3)	5 (16.1)	
Obese	14 (34.2)	15 (15.6)	4 (12.9)	
**Parents’ Weight status**				
Normal weight	13 (33.3)	42 (44.2)	13 (50.0)	0.33
Overweight	13 (33.3)	32 (33.7)	10 (38.5)	
Obese	13 (33.3)	21 (22.1)	3 (11.5)	
	**Median (IQR)** **Mean (SD)**	**Median (IQR)** **Mean (SD)**	**Median (IQR)** **Mean (SD)**	
BMI percentile	87.06 (68.73–95.81)	82.28 (57.35–92.81)	62.45 (34.42–85.89)	
	75.24 (28.16)	74.18 (24.44)	57.44 (33.66)	
Parental BMI	28.05 (22.31–32.77)	25.82 (22.96–29.68)	24.99 (21.92–27.96)	
	28.16 (6.64)	27.47 (7.10)	26.09 (5.47)	

**Table 8 ijerph-20-01382-t008:** Relationship between weight status and parenting styles, food parenting practices, and family meal frequency.

	Number of Times per Week Families Ate Together			*p*-Value
	≤2 Times	3–6 Times	≥7 Times	
	Median (IQR) Mean (SD)	Median (IQR) Mean (SD)	Median (IQR) Mean (SD)	
**Parenting styles**				
Authoritative	3.75 (3.25–4.58)	4.25 (3.50–4.75)	4.75 (3.92–5)	0.0004
	3.87 (0.79)	4.46 (0.64)	4.46 (0.64)	
Authoritarian	3.41 (3.0–4.33)	3.42 (2.91–4.16)	3.33 (2.25–4.50)	0.49
	3.57 (0.90)	3.31 (1.15)	3.32 (1.15)	
Setting rules	4.14 (3.28–5.0)	4.28 (3.57–4.71)	4.57 (3.85–4.86)	0.20
	4.01 (0.88)	4.29 (0.80)	4.29 (0.80)	
Neglecting	3.67 (3.0–4.33)	3.67 (3.0–4.33)	4.33 (3.0–5.0)	0.32
	3.54 (1.15)	3.78 (1.26)	3.79 (1.26)	
**Parenting practices**				
Monitoring	3.0 (2.70–3.80)	3.30 (2.9–3.9)	4.00 (3.30–4.80)	0.0002
	3.19 (0.87)	3.38 (0.79)	3.93 (0.86)	
Reasoning	3.14 (2.71–4.0)	3.42 (2.86–3.86)	4.00 (3.14–4.71)	0.0017
	3.27 (0.87)	3.40 (0.80)	3.89 (0.95)	
Copying	3.33 (2.83–4.0)	3.50 (3.0–4.0)	3.50 (2.83–4.33)	0.75
	3.38 (0.78)	3.40 (0.79)	3.52 (0.97)	
Modeling	3.40 (3.0–4.0)	3.60 (3.2–4.0)	4.00 (3.60–4.80)	0.0008
	3.42 (0.78)	3.57 (0.77)	4.01 (0.91)	

## Data Availability

Data used during the current study are available from the corresponding author.

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
