# Peer review of "Parenting Styles, Food Parenting Practices, Family Meals, and Weight Status of African American Families"

_ijerph, 2023, doi:10.3390/ijerph20021382_

Round 1

Reviewer 1 Report

This is an interesting and well-written article. It serves as an important reminder that eating habits developed during childhood and adolescence can continue throughout an individual's lifetime. The health consequences of obesity are now well-known and the effort to participate in improving eating habits is appreciated. The focus on African American families is a critical step toward filling the gap in our knowledge.

211 African American parent-child dyads make for a robust sample. It is very positive that the data collection was completed within the last year. I am not a statistician and I hope another reviewer can be helpful here.

The study affirms that family meals are associated with with a lower BMI in adolescents, in addition to other positive attributes, including the more positive Authoritative parenting style.

In the list of Authoritarian traits on page 5, #6 on the list, "quality" should be replaced with "guilty"

My concern is that the authors do not address the role of family income in eating habits and risk of obesity. We know that obesity is often related to low socioeconomic status in the US. This issue is mentioned very briefly on page 2 at the end of the second full paragraph. In Table 1 the authors show family income for the sample, yet they do not address this in the Discussion section. I hope the authors will find it worthwhile to include another paragraph to address this.

I'm not convinced that the use of the BMI should really qualify as a limitation of the study. This is a generally used measure but I agree it is certainly important to state that there are other more accurate measures.

The conclusion could be strengthened by including a reminder of the importance of making available fresh fruits and vegetables to all, regardless of race, ethnicity and/or socioeconomic status.

Overall, a very good effort!

Author Response

Uploaded document.

Reviewer 2 Report

The authors investigate the correlation between (i) different parental influences, namely parenting styles, food parenting practices and family meal frequency with weight status among African Americans; and (ii) the correlation between parenting styles and food parenting practices with family meal frequency. They find PSs and FPPs are associated with family meal frequency, and, in turn, meal frequency is associated with ensuring a healthy weight status in adolescents.

The article is relevant and timely. The results are interesting and could be useful for the development of parental education workshops or sessions to help parents adopt appropriate strategies. Methods are appropriate but should be described in more detail to be fully understood. The presentation of results could be improved (see below). The paper could be improved by addressing thoroughly the following comments and suggestions:

Broad comments:

·       The aim and contribution need to be better outlined. Why is it important to conduct the study? Why is it important to conduct the study in a minority group and in this specific minority group? What is new in relation to existing studies? What do the findings add to current knowledge?

·       More detail is needed when describing the methods:

o   The authors state that “A total of 211 African American parent–adolescent dyads participated in this cross-sectional study”. I would suggest discussing the sample size.

o   I would suggest briefly explaining what “W-3003 Multistate Project” and “Comprehensive General Parenting Questionnaire (CGPQ)” are.

o   BMI categories: I would suggest indicating which self-reported anthropometric measurements were collected and how the weight status was generated alongside the sentence “The parents self-reported anthropometric measurements for themselves and their children”. I would also suggest indicating the different two exact types of measures of BMI used for parents and adolescents as used in the correlation analysis and why different types of measures are necessary.

o   The section on Parents’ survey does not include information on how the items used to generate the variables “Parenting Styles” and “Food Parenting Practices” were collected.

o   The correlation tests conducted between certain variables have not been included (correlations with family meal frequency).

o   Regarding the BMI percentile, has the BMI been used as categorical or as continuous variable in the correlation tests reported in Tables 5 and 6?

·     Presentation of results: The authors outline two aims. They aim to find the correlation between (i) different parental influences, namely parenting styles, food parenting practices and family meal frequency with weight status among African Americans; and (ii) the correlation between parenting styles and food parenting practices with family meal frequency. But when presenting the results, I have several issues:

o   First, some of the results have not been reported.

o   Second, it seems that the ordering of reporting the results related to the correlation tests does not follow the order set out in the objectives. I would suggest improving the section’s structure.

Specific comments:

·       P 1: I am not fully convinced the following statement is true “it aims to examine which PSs and FPPs result in a greater frequency of family meals”. I am not fully convinced it is a causal relationship but a correlation. I would suggest somewhat rewriting the sentence to make this clear.

·       P 1: The text reads “No correlation was found between the adolescents’ and parents’ obesity status and the PSs and FPPs, while the adolescents’ BMI percentile was significantly correlated with parental BMI”. I cannot see anywhere in the text the results on the correlation between the parents’ weight status on the one hand and the PSs and FPPs on the other, nor the correlation between adolescents’ BMI percentile and parental BMI. I would suggest including them.

·       P 1: The results to which the following sentence refers give mixed results: “a higher number of family meals per week significantly decreased the likelihood of obesity among the adolescent”. A higher number of family meals decreases the likelihood of obesity among adolescents to some extent and depends on the type of BMI used.

·       P 1: Which National Center for Health Statistics? I would suggest including this information.

·       P 4, Table 1: One of the categories of “Meals together” is wrong.

·       P 7: There is something wrong with subtitle “3.3. PSs, FPPs, and Family Meal Frequency with Weight Status of the Adolescents and Parents”. I would suggest correcting it.

·       P 8: “3.4. Family Meal Frequency and PSs and FPPs” not complete. What about BMI categories?

·       P 8: The table is entitled “Table 6. Relationship between weight status and parenting styles, food-parenting practices, and family meal frequency”, but the table does not show this. It shows the relationship between family meal frequency and the other three variables.

·       P 10: The text reads “while monitoring, reasoning, and modelling practices were effective in improving the frequency of family meals”. It is not a causal relationship but a correlation. I would suggest somewhat rewriting the sentence to make this clear.

Author Response

Uploaded document.

Round 2

Reviewer 2 Report

The paper has been improved and most of the comments have been addressed satisfactorily. I have no more comments.